# Effects of α-Naphthylacetic Acid on Cadmium Stress and Related Factors of Tomato by Regulation of Gene Expression

Xiaoxi Guan [1],*, Changling Sui [1], Kecui Luo [1], Zhifeng Chen [1], Chaoyang Feng [1], Xiufen Dong [2], Boping Zeng [1], Xian Dong [1] and Xiaofang Liu [1]

[1] College of Biology and Agricultural Science and Technology, Zunyi Normal University, Zunyi 563006, China
[2] College of Horticultural Sciences, Zhejiang Agriculture and Forestry University, Hangzhou 310000, China
* Correspondence: guanxiaoxi1208@163.com

**Abstract:** Cadmium (Cd) is absorbed and accumulated by crops, and it adversely affects plant growth and development. To explore the effect of exogenous auxin on Cd stress, we applied different concentrations of α-naphthaleneacetic acid (NAA) and the auxin transport inhibitor 2,3,5-triiodobenzoic acid (TIBA) to tomato plants exposed to Cd stress in a hydroponic system. NAA and TIBA at different concentrations were used under Cd stress. Plant growth, root morphology, and auxin distribution were observed. Lipid peroxidation and antioxidant enzyme activities in leaves, cadmiumcontent, and migration coefficient of plants were determined. Transcriptome sequencing and qRT-PCR were used to analyze the differentially expressed genes. Results showed that auxin was concentrated in the leaf veins, stem base, and roots in P5::GUS "Chico III" transgenic tomato, indicating NAA polar transport to the roots and promotion of root growth under Cd stress. Cd was absorbed by the roots and transported to the shoots. It then inhibited plant growth and promoted antioxidant enzyme activities, $O_2^-$ production, $H_2O_2$ accumulation, and membrane lipid peroxidation. Treatment with 0.5 μM NAA improved antioxidant enzyme activities, reduced reactive oxygen, maintained membrane permeability, and decreased malondialdehyde and proline contents. Transcriptome analysis revealed that NAA activated a large number of genes in the roots: 1998 genes were differentially expressed in response to Cd or NAA treatment, and 1736 genes were specifically expressed in response to NAA treatment under Cd stress. Among the differentially expressed genes, tomato metallocarboxypeptidase inhibitor *TCMP*-2 (*2A11*) and *Solanum lycopersicum* heavy metal-associated isoprenylated plant protein (HIPP) 7-like (*LOC101264884*), which are closely related to plant response to heavy metal stress, may be the key sites of NAA. In conclusion, the NAA-mediated response to Cd stress was closely associated with "defense response" genes in shoots and "oxidoreductase activity, oxidizing metal ions" and "response to auxin" genes in roots.

**Keywords:** *Solanum lycopersicum*; α-naphthaleneacetic acid; cadmium stress; gene expression

## 1. Introduction

Heavy metal contamination of soil, especially caused by mining and industrial activities, poses a serious risk to agricultural production and product safety and is an urgent problem to address [1]. Cadmium (Cd) is a common heavy metal pollutant in vegetable crops. As a non-essential element, it can be absorbed and accumulated in various tissues by plants. Moreover, it disrupts normal physiological and biochemical processes and has adverse impacts on the growth and development of plants. In recent years, the mechanism and regulatory pathway of Cd tolerance in plants have received research attention [2]. Studies on the distribution of nutrient elements in different parts of tomato showed that Cd affects the absorption and transportation of certain other elements essential for plant growth, such as zinc and iron [3]. Tomato (*Solanum lycopersicum* L.) is the predominant vegetable produced in protected cultivation, and its growth and development are readily affected by Cd from many aspects. Cd stress can lead to various toxicity symptoms, such as





growth arrest. Moreover, the cell structure can be disturbed, and the activities of ascorbate peroxidase and glutathione reductase decrease significantly in the ascorbate–glutathione (GSH) cycle of tomato roots [4]. Under suitable conditions, the antioxidant enzyme system can eliminate the reactive oxygen radicals induced by stress, and alleviate the damage caused by stress to a certain extent. SOD firstly disproportionates reactive oxygen radicals and then converts them into $H_2O_2$ which can be cleared by CAT and POD. Various environmental conditions lead to a large number of reactive oxygen species and free radicals in plants. $H_2O_2$ and $O_2^-$ are the most important reactive oxygen species in plants, which lead to membrane lipid peroxidation and indirectly reflect the degree of stress. Studies have shown that many foreign substances can reduce the accumulation of ROS in plants and enhance tolerance to Cd stress [5]. Proline is an important osmoregulatory substance that is readily induced by stress. As one of the most important products of membrane lipid peroxidation, MDA is a marker of cell membrane system damage [6].

In recent years, many studies have examined the mechanism of Cd tolerance in tomato, of which the majority have focused on the evaluation of stress tolerance and stress mitigation [7,8]. Nogueirol et al. showed that different nitrogen sources (ammonium- and nitrate-nitrogen) differentially alleviate Cd stress in tomato [9]. Wang et al. analyzed the metabolism of metallothionein and GSH-derived phytochelatins under Cd and copper stress and confirmed the role of nitric oxide in the heavy metal tolerance of tomato [10]. Following the treatment of tomato plants with 5 mM GSH under Cd stress, the contents of nitric oxide and *S*-nitrosothiols are increased, the redox state is adjusted, the expression of associated transcription factors and stress-responsive genes is altered, and the stress tolerance of the plant is improved [11]. At present, the application of exogenous substances to improve plant Cd stress resistance is an important research focus. By observing the phenotype of auxin and ethylene mutants under Cd stress; measuring the contents of malondialdehyde (MDA), hydrogen peroxide ($H_2O_2$), and soluble proteins; and evaluating the role of Cd accumulation and distribution in the mutants, the coordination of auxin and ethylene in tomato plants under Cd stress was revealed [12].

Previous studies revealed the response mechanism of auxin-related genes to a variety of environmental factors [13–16]. A previous study confirmed that auxin migration affects the tolerance of plants to salt and high-temperature stress, and the constructed *aux1* mutant is significantly sensitive to temperature and salt stress, proving that the *AUX1* gene is involved in the regulation of stress tolerance [17]. Family identification and expression profile analysis of auxin transport-related genes in maize showed that most *AUX/LAX*, *PIN*, and *ABCB* genes are induced by drought, salt, and low temperature [18]. Expression profile analysis confirmed that most ramie *Aux/IAA* and *ARF* gene families have differential expression patterns under drought and high-temperature stress [19]. Overexpression of the wheat *SAUR78* gene in *Arabidopsis* can reduce intracellular $H_2O_2$ content and improve plant resistance to drought, low temperature, and salt stress. The expression of related stress resistance genes is up-regulated at the same time, which proves that *SAUR78* is closely related to a variety of abiotic stresses [20]. The researchers confirmed that some *DREB/CBF TF* transcription factors directly induce *IAA19* and *IAA5* to participate in the abiotic stress response in *Arabidopsis* [21]. Auxin signals can mediate the multi-factor response regulation process, which proves that its response to abiotic stress is pleiotropic. However, the mechanism of auxin-mediated alleviation of Cd stress in tomato growth and development has not been reported previously. In auxin substances, indoleacetic acid (IAA) and indole butyrate (IBA) are easy to decompose in light and air, and auxin analogue (2,4-D) has high biological toxicity. By contrast, naphthylacetic acid (NAA) has stable chemical property and low toxicity.

Therefore, in this study NAA was selected to explore the role of auxin in alleviating and reducing Cd stress on tomato growth. NAA and 2,3,5-triiodobenzoic acid (TIBA) were applied to observe tomato plant growth and root morphology; detect the distribution of plant $Cd^{2+}$ content; and analyze the antioxidant mechanism, endogenous auxin level, and temporal and spatial changes in this study. Transcriptome sequencing was used to

screen the differentially expressed genes (DEGs), screen and verify the key genes of the auxin signaling pathway, and clarify the physiological and molecular mechanism of auxin-mediated Cd tolerance in tomato. This study provides a theoretical basis for the genetic resistance breeding of solanaceous vegetables represented by tomato.

## 2. Materials and Methods

### 2.1. Materials

Tomato (*Solanum lycopersicum* L.) 'Zhongshu No. 6' was provided by the Chinese Academy of Agricultural Sciences, Beijing. The transgenic tomato P5::GUS 'Chico III' was from the C.M. Rick Tomato Genetics Resource Center (TGRC), in which the auxin inducible promoter of the GUS reporter gene can visually locate auxin and be applied to the study of auxin dynamics in the process of tomato growth [22]. NAA and the auxin transport inhibitor TIBA were provided by Macklin Biochemical Technology Co., Ltd. (Shanghai, China). All other reagents were of AR grade.

### 2.2. Treatments

The seeds were soaked in warm water, sterilized, and germinated onfilter paper. After the seedlings emerged, they were irrigated with quarter-strength Hoagland's nutrient solution. When the third true leaf unfolded, plants of uniform growth were selected, and their roots were washed and transferred to half-strength Hoagland's nutrient solution. After 1 week of pre-culture, the nutrient solution was replaced with full-strength Hoagland's nutrient solution. The seedlings were cultured at 14 h/10 h (day/night) photoperiod, 25 °C/20 °C (day/night), and 80% relative humidity and then supplied with continuous ventilation. The nutrient solution was replaced at weekly intervals. Treatments started every other day when the fourth true leaf unfolded.

To determine the role of exogenous auxin in Cd stress, 'Zhongshu No. 6' seedlings were treated with NAA at different concentrations. TIBA, the specific inhibitor of auxin to block the polar transport of auxin, was supplied to verify the role of NAA. The treatments were as follows: control (C, distilled water), Cd (50 μM), Cd + N1 (50 μM Cd + 0.5 μM NAA), Cd + N2 (50 μM Cd + 1 μM NAA), Cd + T1 (50 μM Cd + 100 μM TIBA), and Cd + T2 (50 μM Cd + 200 μM TIBA), with 150 seedlings per treatment. Cd stress was induced by treatment with Cd-containing nutrient solution; NAA and TIBA were applied by foliar spray of solutions. The seedlings were supplied with continuous ventilation, and the nutrient solution was replaced at weekly intervals.

### 2.3. Distribution of Auxin in the Plant and Determination of Growth

The P5::GUS transgenic tomato plants transformed with an auxin-inducible promoter used to detect the distribution of auxin. After 2 weeks of treatment, the harvested plant samples were cut off at the stem base of the plant and wiped dry with filter paper. To explore the effect of auxin, four treatments (C, Cd, Cd + N1, and Cd + T1) were stained with the GUS staining kit (Beijing Coolaber Company, Beijing, China). The growth indices of all treatment groups were measured after the appearance of the sixth true leaf. Plant height was the vertical distance from the base of the root to the highest point of the tomato. Vernier caliper was used to measure the stem diameter between the second and third leaves of tomato. The root length was measured from the base to the tip. After root stem separation and the water had been wiped dry, the fresh weight was measured directly with an analytical balance. Dry weight was performed first in the oven with 105 °C for 0.5 h and then in 80 °C drying to constant weight. SPAD-502 PLUS chlorophyll meter (Konica Minolta Sensing, Osaka, Japan) was used to estimate leaf chlorophyll content [23].

### 2.4. Determination of MDA, Proline and $H_2O_2$ Content, Superoxide Anion Production Rate, and Antioxidant Enzyme Activity

The MDA content was determined according to the color reaction of thiobarbituric acid (TBA) [24]. The proline content was estimated using the Coomassie brilliant blue

G-250 method [25]. The superoxide anion production rate was detected using the method of hydroxylamine oxidation described by Elstner and Heupal [26]. The $H_2O_2$ content was determined using a hydrogen peroxide kit (Nanjing Jiancheng Bioengineering Institute, Nanjing, China). A fresh leaf sample (0.5 g) was homogenized in 0.1 M PBS. After centrifugation, the antioxidant enzyme crude extract was dissolved in supernatant. Superoxide dismutase (SOD, EC1.15.1.1) activity was measured according to Beyer and Fridovich [27]. Peroxidase (POD, EC 1.11.1.7) and catalase (CAT, EC 1.11.1.6) activities were measured using the methods described by Chance and Maehly [28].

### 2.5. Determination of Cd Content and Transport Coefficient

The shoots and roots of the harvested plant samples were separated, dried at 105 °C to constant weight, and digested with nitric acid-perchloric acid. The Cd content was determined using an atomic absorption spectrophotometer (AA-7000, Kyoto, Japan). The translocation factor (TF) was calculated as TF = $Cd^{2+}$ content in the shoots/$Cd^{2+}$ content in roots [29].

### 2.6. RNA-Seq Transcriptome Analysis

After 2 weeks of treatment, the harvested plant samples were cut off at the stem base of the plant. The shoot and root samples of the control group were designated A1_1 and A1_2, respectively. After treatment, the shoot samples were designated A2_1 (Cd) and A3_1 (Cd + N1), whereas the root samples were designated A2_2 (Cd) and A3_2 (Cd + N1). Each treatment had three biological replicates for RNA-Seq analysis. The samples were frozen in liquid nitrogen and stored at −80 °C until RNA isolation [30]. The corresponding samples were retained for further qRT-PCR detection and verification. The total RNA Extraction Kit (Tiangen, Beijing, China) was used in RNA extraction. The RNA samples were reverse-transcribed into cDNA and subjected to transcriptome sequencing by Nuohe Zhiyuan Biotechnology Co., Ltd. (Beijing, China). Shoot and root samples were used to prepare RNA libraries for RNA sequencing analysis (Table S1). DEGs were identified based on the criteria ($p < 0.05$; $|\log_2$fold change$| > 1.0$). The identified DEGs were subjected to gene ontology (GO) and the Kyoto Encyclopedia of Genes and Genomes (KEGG) enrichment analyses [31].

### 2.7. qRT-PCR Analysis

The expression levels of *LOC101245781*, *LOC101267701*, *LOC101266326*, *CYP707A1*, *LOC101258738*, *IPT3*, *LOC101262425*, *Arf/Xyl1*, *LOC101253212*, *LOC101259471*, *LOC101248212*, and *LOC101245398* were analyzed by qRT-PCR, with *Actin* as internal control (Table S2) [32]. In brief, the cDNAs were combined with each primer and Talent qPCR PreMix (SYBR Green) (Tiangen, Beijing, China) for qRT-PCR analysis by Sequence Detection System (ABI Prism 7500, New York, NY, USA). The reaction procedure was as follows: initial denaturation at 95 °C for 3 min, 40 cycles of denaturation at 95 °C for 5 s, and annealing/extension at 60 °C for 15 s. Specific primer pairs are displayed in Table S1. The relative mRNA levels were quantified against that of the internal standard gene.

### 2.8. Statistical Analysis

The data were subjected to ANOVA, and the significance of differences was analyzed by multiple comparisons ($p < 0.05$).

## 3. Results

### 3.1. Auxin Distribution and Growth of Tomato Plants under Cd Stress

In the untreated control, auxin was mainly concentrated in the growth points, below the nodes of the first and second true leaves, and in the root hair zone (Figure 1). Compared with the control, the growth of shoots and roots under Cd treatment was delayed and hindered to a certain extent. The roots were sparse and weak, and auxin accumulation decreased in the root system. However, Cd + N1 treatment restored the shoot and root

growth potential to a certain extent. Auxin was distributed in the leaf veins and accumulated in the base of the main stem and in the root system. By contrast, the plant showed conspicuous dwarfism under Cd + T1 treatment. The main stem was bent, the lateral shoot in the axil of the first true leaf developed, and auxin was especially concentrated around the growth points. Transverse sections of the primary root base revealed that auxin was mainly concentrated in the pith of the primary root under the different treatments, and it gradually decreased in concentration radially from the pith to the pericycle, endodermis, and exodermis. Cd + N1 treatment promoted the accumulation of auxin in lateral roots. These results showed that foliar application of NAA promoted the growth of the root system under cadmium stress. The polar transport of auxin was restored in tomato plants to a certain extent. Under treatment with TIBA, auxin was not transported readily and was concentrated in the terminal portion of the shoot, which strongly aggravated the adverse effects of Cd stress on plant growth.

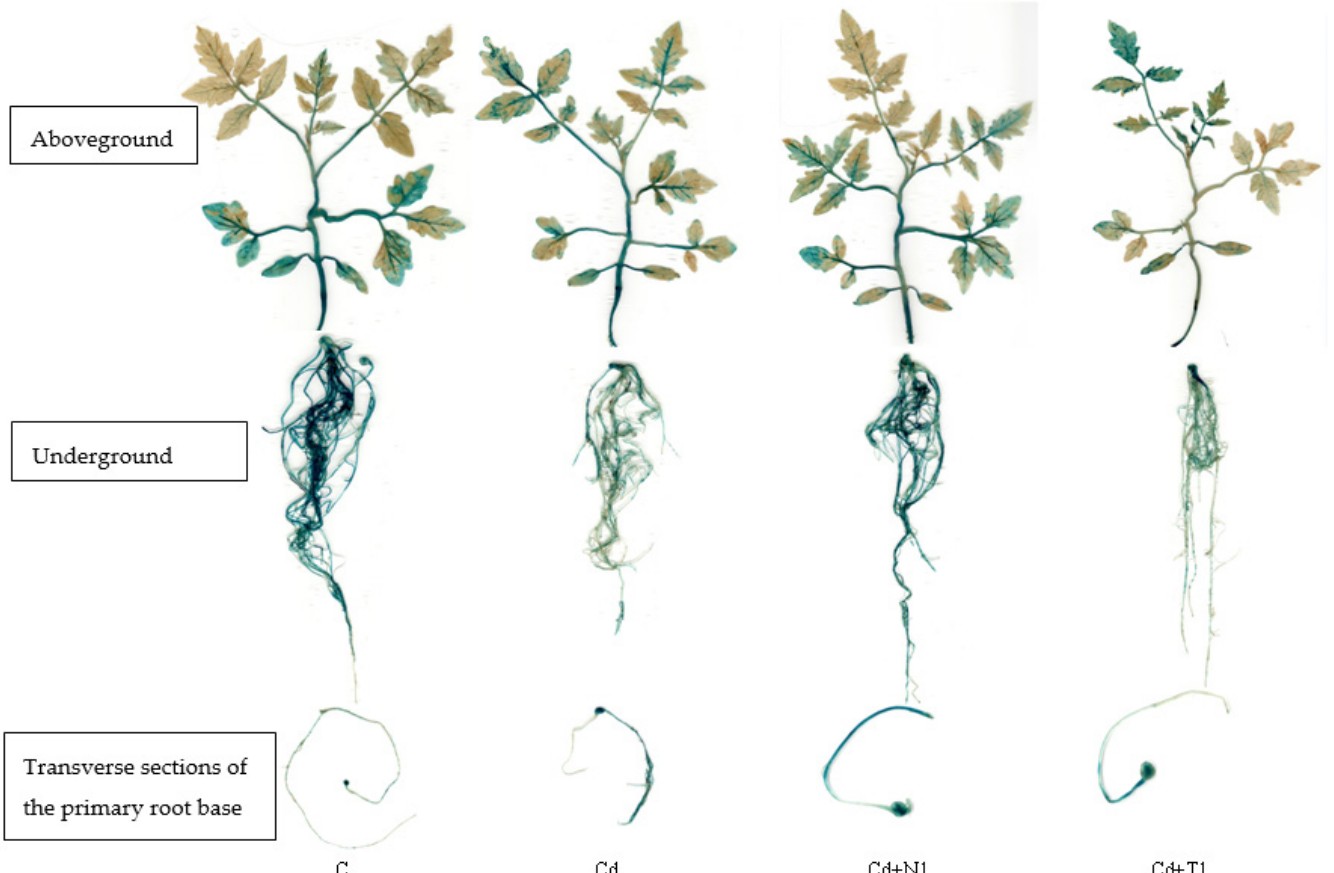

**Figure 1.** Distribution of auxin in tomato plants under Cd stress.C: 0 μM Cd; Cd: 50 μM Cd; Cd + N1: 50 μM Cd + 0.5 μM NAA; Cd + N2: 50 μM Cd + 1 μM NAA; Cd + T1: 50 μM Cd + 100 μM TIBA; Cd + T2: 50 μM Cd + 200 μM TIBA.

The plant height, stem diameter, SPAD value of leaves, and primary root length significantly decreased under Cd treatment, which was highly significant except for the primary root length, with percentage decreases of 17.3%, 26.5%, 27%, and 11.7%, respectively (Table 1). Cd + N1 and Cd + N2 treatments significantly increased the plant height and leaf SPAD value compared with Cd treatment. Cd + N1 treatment significantly increased the primary root length, whereas Cd + N2 treatment significantly increased the stem diameter compared with Cd treatment by 35.3%. Cd + T1 and Cd + T2 treatments significantly reduced the primary root length and leaf SPAD value compared with those under Cd treatment. Cd + T2 treatment significantly inhibited height growth by 15.4% compared with Cd treatment. The fresh weight and dry weight of shoots and roots of

tomato plants decreased significantly under Cd treatment (Table 1). Cd + N1 and Cd + N2 treatments significantly increased the shoot biomass compared with Cd treatment. Cd + N1 treatment significantly increased the fresh weight and dry weight of roots by 18.5% and 19.9%, respectively (Table 2). Cd + T1 and Cd + T2 treatments significantly reduced the fresh weight of roots compared with that under Cd stress. Cd + T2 treatment significantly reduced the dry weight of shoots and roots by 14% and 17.1%, respectively. These results showed that Cd stress inhibited the growth of tomato plants, and NAA treatment alleviated this effect. The regulatory effect of 0.5 μM NAA (Cd + N1) was more effective than that of 1.0 μM NAA (Cd + N2). The polar auxin transport inhibitor TIBA intensified the growth inhibition induced by Cd stress, especially in the roots. The effect was intensified with the increase in TIBA concentration (200 μM vs. 100 μM). The application of NAA promoted the growth of tomato plants under cadmium stress, TIBA treatment proved this result from the opposite side.

**Table 1.** Growth indices of tomato plants under Cd stress.

| Treatments | Plant Heigh (cm) | Stem Diameter (mm) | Main Root Length (cm) | SPAD Value |
|---|---|---|---|---|
| C | 22.73 ± 0.40 [a] | 4.12 ± 0.16 [a] | 32.60 ± 0.87 [a] | 40.07 ± 0.83 [a] |
| Cd | 18.80 ± 0.26 [b] | 3.03 ± 0.03 [bc] | 28.80 ± 1.08 [b] | 29.27 ± 1.33 [c] |
| Cd + N1 | 21.33 ± 1.15 [a] | 3.19 ± 0.07 [b] | 33.10 ± 1.05 [a] | 33.73 ± 1.04 [b] |
| Cd + N2 | 22.47 ± 1.29 [a] | 4.10 ± 0.06 [a] | 26.80 ± 3.48 [bc] | 36.13 ± 2.60 [b] |
| Cd + T1 | 17.50 ± 0.50 [b] | 3.06 ± 0.04 [bc] | 24.13 ± 2.50 [c] | 25.67 ± 1.53 [d] |
| Cd + T2 | 15.90 ± 0.66 [c] | 2.95 ± 0.19 [c] | 24.60 ± 1.42 [c] | 22.60 ± 1.18 [e] |

The lowercase letters indicate the significant differences at 5% level. C: 0 μM Cd; Cd: 50 μM Cd; Cd + N1: 50 μM Cd + 0.5 μM NAA; Cd + N2: 50 μM Cd + 1 μM NAA; Cd + T1: 50 μM Cd + 100 μM TIBA; Cd + T2: 50 μM Cd + 200 μM TIBA.SPAD value: the relative value of chlorophyll content in leaves.

**Table 2.** Fresh and dry weight of tomato plants under Cd stress.

| Treatments | Fresh Weight (g) | | Dry Weight (mg) | |
|---|---|---|---|---|
| | Shoots | Roots | Shoots | Roots |
| C | 5.20 ± 0.27 [a] | 0.80 ± 0.03 [a] | 440 ± 17 [a] | 44.60 ± 3.27 [a] |
| Cd | 3.58 ± 0.16 [c] | 0.54 ± 0.02 [c] | 299 ± 6 [c] | 36.07 ± 2.58 [bc] |
| Cd + N1 | 4.35 ± 0.12 [b] | 0.64 ± 0.05 [b] | 364 ± 34 [b] | 43.23 ± 5.29 [a] |
| Cd + N2 | 4.78 ± 0.30 [ab] | 0.56 ± 0.04 [c] | 429 ± 23 [a] | 40.30 ± 1.31 [ab] |
| Cd + T1 | 3.41 ± 0.43 [c] | 0.44 ± 0.03 [d] | 300 ± 14 [c] | 32.43 ± 2.28 [cd] |
| Cd + T2 | 3.10 ± 0.14 [c] | 0.34 ± 0.04 [e] | 257 ± 18 [d] | 29.90 ± 1.21 [d] |

The lowercase letters indicate the significant differences at 5% level. C: 0 μM Cd; Cd: 50 μM Cd; Cd + N1: 50 μM Cd + 0.5 μM NAA; Cd + N2: 50 μM Cd + 1 μM NAA; Cd + T1: 50 μM Cd + 100 μM TIBA; Cd + T2: 50 μM Cd + 200 μM TIBA.

*3.2. MDA and Proline Contents, $O_2^-$ Production Rate, and $H_2O_2$Content of Tomato Plants under Cd Stress*

The leaf MDA content of tomato plants under Cd treatment significantly increased by 54.5%, that under Cd + N1 treatment significantly decreased by 28.4%, and that under Cd + T2 treatment significantly increased (Figure 2A,B). The leaf proline content increased significantly under Cd treatment; decreased by 24% and 43.4% under Cd + N1 and Cd + N2 treatments, respectively; and increased significantly under Cd + T2 treatment. The contents of MDA and proline increased under cadmium stress, while foliar application of NAA effectively reduced the accumulation of these substances.

Many adverse environmental conditions lead to the accumulation of reactive oxygen species (ROS) and free radicals in plants. Superoxide anion ($O_2^-$) and $H_2O_2$ are the most important ROS in plants, which can lead to membrane lipid peroxidation. Their contents indirectly reflect the degree of stress. Cd treatment significantly increased the $O_2^-$ production rate and $H_2O_2$ content in the leaves of tomato plants, whereas Cd + N1 and Cd + N2 treatments significantly decreased the $O_2^-$ production rate by 21.2% and 47.7%,

respectively (Figure 2C,D). Compared with Cd treatment, the $H_2O_2$ content under Cd + N1 treatment significantly decreased by 24.4%. These results showed that Cd stress promoted $O_2^-$ production and $H_2O_2$ accumulation to a certain extent, which was not conducive to the maintenance of a stable cell structure. The 0.5 μM NAA treatment effectively inhibited the production and accumulation of these substances, which helped in the recovery of a normal cell structure.

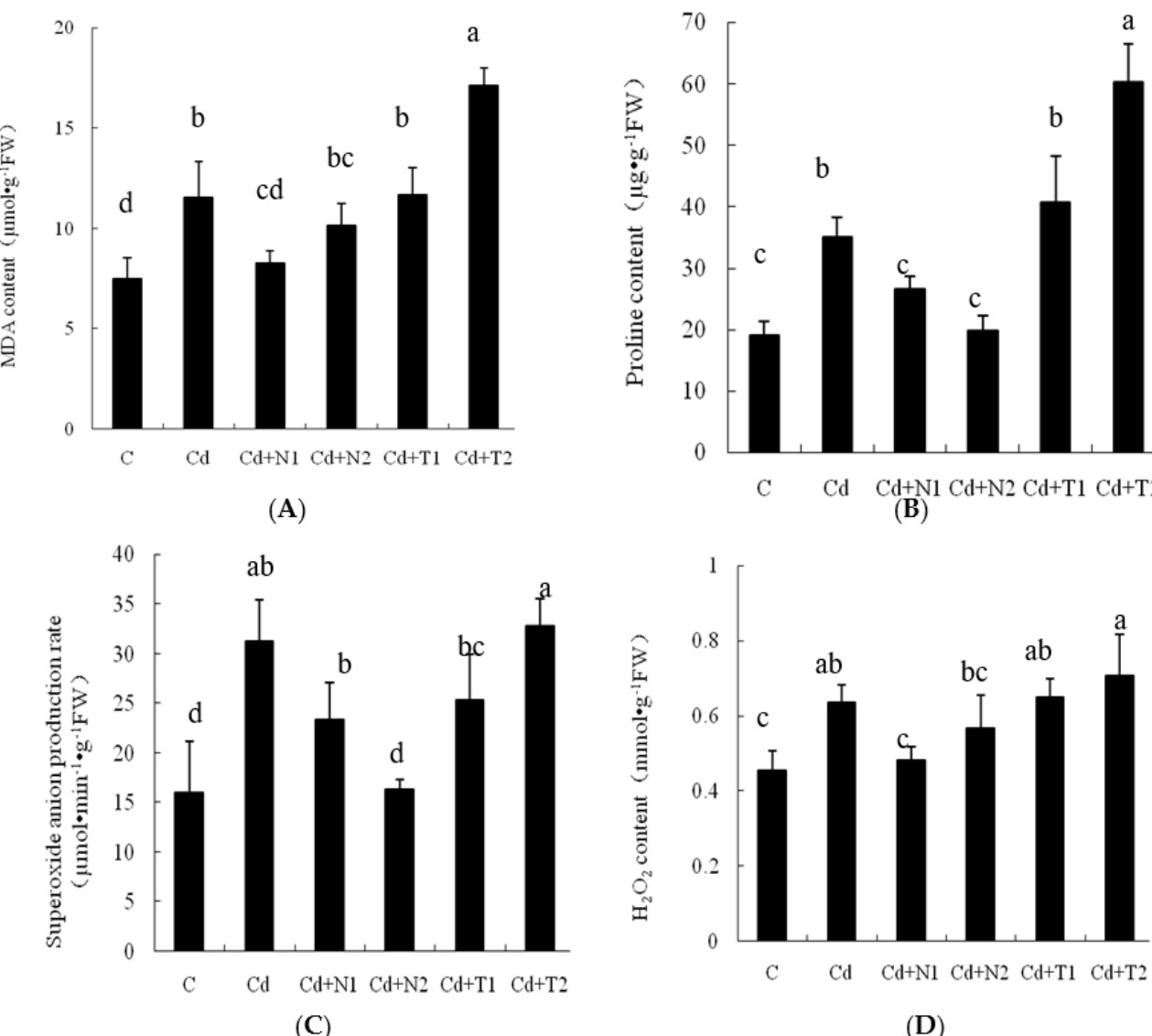

**Figure 2.** MDA contents (**A**), proline contents (**B**), superoxide anion ($O_2^-$) production rate (**C**), and $H_2O_2$ (**D**) content in leaves of tomato plants under Cd stress.C: 0 μMCd; Cd: 50 μMCd; Cd + N1: 50 μM Cd + 0.5 μM NAA; Cd + N2: 50 μM Cd + 1 μM NAA; Cd + T1: 50 μM Cd + 100 μM TIBA; Cd + T2: 50 μM Cd + 200 μM TIBA; MDA: malondialdehyde; $H_2O_2$: hydrogen peroxide; FW: fresh weight. The lowercase letters indicate the significant differences at 5% level.

### 3.3. Antioxidant Enzyme Activities of Tomato Plants under Cd Stress

Activities of SOD, POD, and CAT in the leaves increased under Cd treatment by 21.1%, 64.4%, and 35.6%, respectively (Table 3). Cd + N1 and Cd + N2 treatments significantly increased POD activities by 40.4% and 67.4%, respectively, and increased SOD activity significantly. The 0.5 and 1.0 μM NAA treatments further enhanced the activities of antioxidant defense enzymes, promoted ROS scavenging, and maintained relatively normal physiological activity to a certain extent. Application of TIBA was not conducive to

the functions of the antioxidant defense enzymes. Auxin signaling was involved in the regulation of antioxidant enzyme activities in tomato under Cd stress.

### 3.4. Cd Absorption and Translocation in Tomato Plants under Cd Stress

Cd treatment significantly increased the Cd content of the whole tomato plant, whereas Cd + N1 treatment significantly decreased the Cd content of the shoots and roots by 39.1% and 20.4%, respectively (Figure 3). Cd + N2 treatment significantly decreased the Cd content of shoots by 18.4% but had no significant effect on that of the roots. Cd + T1 and Cd + T2 treatments significantly increased the Cd content of the whole tomato plant. Cd treatment significantly increased Cd transport from the roots to the shoots. Cd + N1 treatment significantly reduced Cd transport by 22.8%, whereas the other treatments had no significant effect. Application of NAA effectively down-regulated Cd uptake in plants under Cd stress. The inhibition on different parts of the plant was associated with the NAA concentration.

**Table 3.** Antioxidant enzyme activities of tomato plants under Cd stress.

| Treatments | SOD Activity (Umin$^{-1}$g$^{-1}$FW) | POD Activity (Umin$^{-1}$g$^{-1}$FW) | CAT Activity (Umin$^{-1}$g$^{-1}$FW) |
|---|---|---|---|
| C | 3.8 ± 0.06 [cd] | 23.3 ± 4.86 [d] | 27.0 ± 2.62 [c] |
| Cd | 4.6 ± 0.24 [bc] | 38.3 ± 1.56 [c] | 36.6 ± 3.60 [b] |
| Cd + N1 | 5.8 ± 0.38 [a] | 53.8 ± 5.62 [ab] | 42.8 ± 5.96 [ab] |
| Cd + N2 | 4.8 ± 0.63 [b] | 64.1 ± 10.88 [a] | 46.7 ± 2.13 [a] |
| Cd + T1 | 3.7 ± 0.63 [d] | 38.0 ± 9.81 [c] | 27.6 ± 2.34 [c] |
| Cd + T2 | 3.6 ± 0.49 [d] | 43.3 ± 8.67 [bc] | 24.1 ± 4.37 [c] |

The lowercase letters indicate the significant differences at 5% level. SOD: superoxide dismutase, POD: peroxidase, CAT: catalase. C: 0 μM Cd; Cd: 50 μM Cd; Cd + N1: 50 μM Cd + 0.5 μM NAA; Cd + N2: 50 μM Cd + 1 μM NAA; Cd + T1: 50 μM Cd + 100 μM TIBA; Cd + T2: 50 μM Cd + 200 μM TIBA.

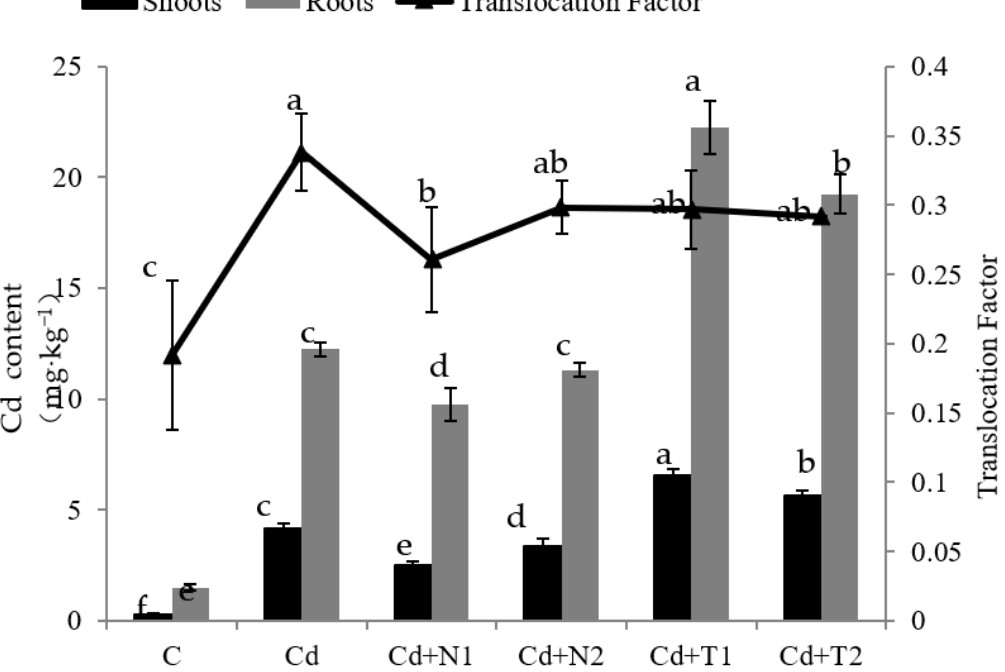

**Figure 3.** Cd uptake and translocation of tomato plants under Cd stress. The lowercase letters indicate the significant differences at 5% level. C: 0 μM Cd; Cd: 50 μM Cd; Cd + N1: 50 μM Cd + 0.5 μM NAA; Cd + N2: 50 μM Cd + 1 μM NAA; Cd + T1: 50 μM Cd + 100 μM TIBA; Cd + T2: 50 μM Cd + 200 μM TIBA.

### 3.5. RNA-Sequencing Analysis

Compared with the control, 1107 DEGs were detected in tomato shoots under Cd treatment, including 367 up-regulated genes and 740 down-regulated genes. Furthermore, 6372 DEGs were detected in the roots under Cd treatment, including 2848 up-regulated genes and 3524 down-regulated genes. Compared with Cd treatment, 358 DEGs were detected in the shoots under Cd + N1 treatment (including 271 up-regulated genes and 87 down-regulated genes) and 3734 DEGs in the roots under Cd + N1 treatment (including 2016 up-regulated genes and 1718 down-regulated genes).

Compared with the control, 383 genes were generally expressed in the shoots and roots under Cd treatment, whereas 724 genes were only expressed in the shoots and 5989 genes were only expressed in the roots (Figure 4). Compared with Cd treatment, 78 genes were universally expressed in the shoots and roots under Cd + N1 treatment, 280 genes were only expressed in the shoots, and 3656 genes were only expressed in the roots. In the shoots, 241 genes were generally expressed under Cd and Cd + N1 treatments, whereas 117 genes were specifically expressed under Cd + N1 treatment. In the roots, 1998 genes were generally expressed under Cd and Cd + N1 treatments, and 1736 genes were specifically expressed under Cd + N1 treatment. Combined with the comparisons A3_1 vs. A2_1 and A3_2vs. A2_2, these findings showed that Cd + N1 treatment induced 61 genes specifically regulated in the shoots, 1615 genes specifically regulated in the roots, and 8 genes regulated in both organs.

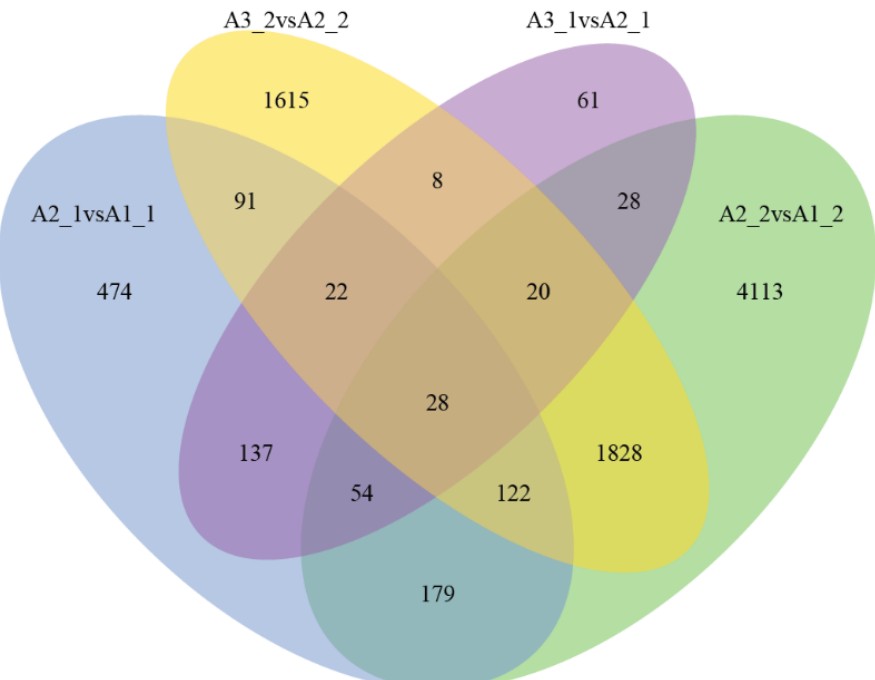

**Figure 4.** "A2_1 vs. A1_1" indicates the comparison of gene expression in the A2_1 library with that in the A1_1 library, and three other comparisons are indicated.

### 3.6. Analysis of Differential Expression Patterns

The combined gene expression patterns were grouped into four major clusters, which were designated Clusters A, B, C, and D (Figure 5). The differential expression of genes in Clusters A, B, and C was mainly concentrated in the roots. The majority of Cluster A genes were up-regulated under Cd treatment and down-regulated under Cd + N1 treatment. Cluster B-1 genes were down-regulated under Cd treatment. Cluster B-2 genes were up-regulated under Cd treatment and down-regulated under Cd + N1 treatment. Cluster C-1 genes were down-regulated under Cd treatment and up-regulated under Cd + N1 treatment. Cluster C-2 genes were down-regulated under Cd and Cd + NAA treatments.

Cluster D-1 genes were up-regulated under Cd treatment and down-regulated slightly under Cd + N1 treatment. Cluster D-2 genes were down-regulated under Cd treatment and up-regulated under Cd + N1 treatment.

*3.7. GO and KEGG Analyses of DEGs Responsive to NAA Treatment*

The DEGs were classified into three categories in GO enrichment analysis, and they were mainly distributed in the following ways (Figure 6A): The DEGs classified in the biological process category were predominantly associated with cell wall organization or biogenesis and polysaccharide metabolic process. The DEGs classified in the cellular component category were largely associated with the extracellular region. In the molecular function category, the majority of DEGS were associated with tetrapyrrole binding, followed by heme binding. Thus, the effect of NAA on tomato under Cd stress was regulated by DEGs in these enriched pathways.

The top 20 pathways with the lowest $p_\text{adj}$ are shown in Figure 6B. The following pathways showed the lowest $p_\text{adj}$ in the roots, including photosynthetic antenna proteins, photosynthesis, phenylpropanoid biosynthesis, zeatin biosynthesis, pentose and glucuronate interconversions, DNA replication, and flavonoid biosynthesis. These pathways were the most enriched with DEGs. Forty DEGs were associated with phenylpropanoid biosynthesis, and the gene ratio was 0.06.

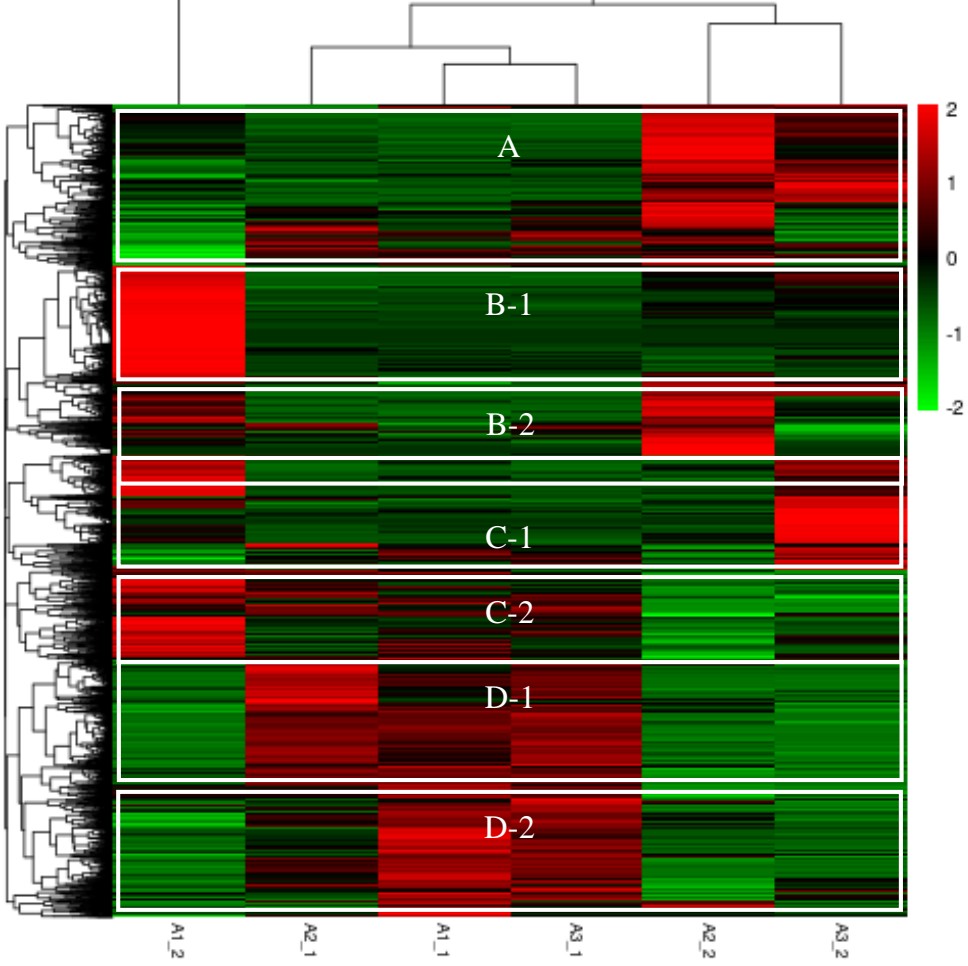

**Figure 5.** Hierarchical cluster analysis of genes differentially expressed in response to exogenous NAA treatment under Cd stress. This map shows the genes of log₂ (fold change) values of the NAA effects on the shoot and root. Colors represent changes in gene expression levels: red means up, and green means down. The combined gene expression patterns were grouped into four major clusters, which were designated Clusters A, B, C, and D.

### 3.8. Verification of Selected DEGs by Quantitative Real-Time PCR

A qRT-PCR of DEGs was performed to verify RNA-Seq data (Figure 7). Among the nine genes, three were up-regulated (*LOC101245781*, *LOC101267701*, and *LOC101266326)* in the comparison A3_1 vs. A2_1; two were up-regulated (*CYP707A1* and *LOC101258738*) and one was down-regulated (*IPT3*) in the comparison A3_2 vs. A2_2; two were up-regulated (*Arf/Xyl1* and *LOC101253212*) and one was down-regulated (*LOC101262425*) in the comparison A2_2 vs. A1_2; and two were up-regulated (*LOC101245398* and *LOC101259471*) and one was down-regulated (*LOC101248212*) in the comparison A2_1 vs. A1_1. The results were consistent with those of RNA-Seq analysis and confirmed that the RNA-seq data were reliable.

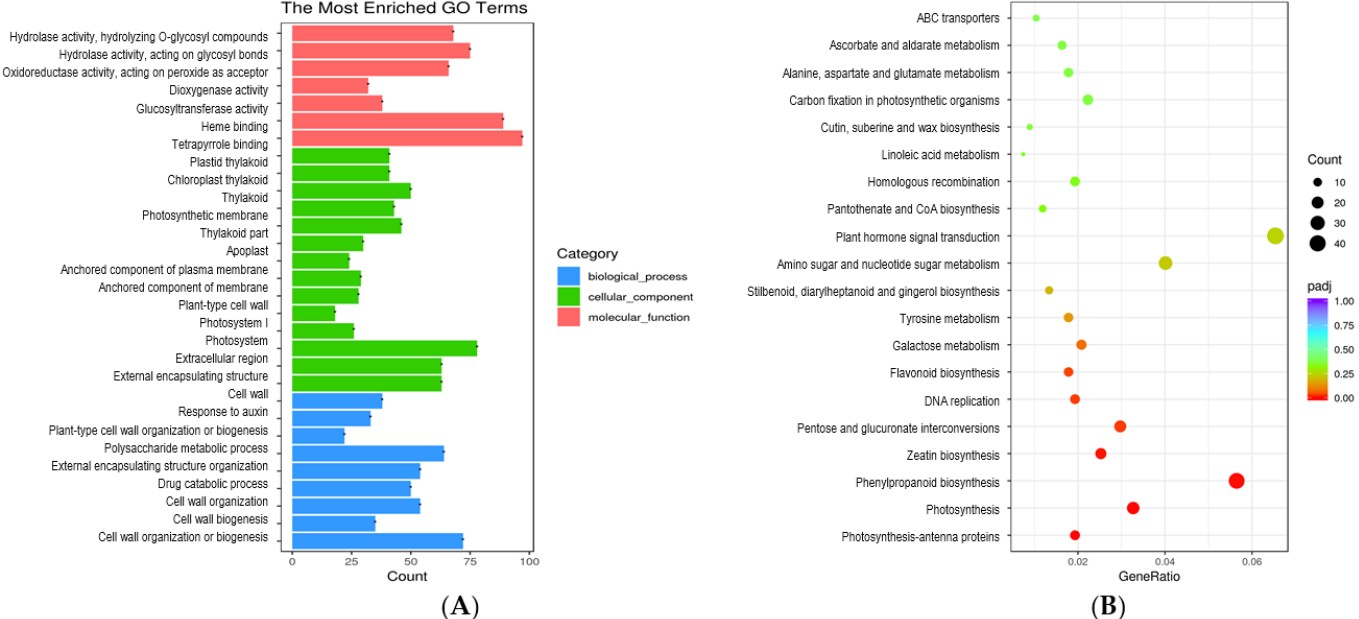

**Figure 6.** Gene ontology (GO) enrichment analysis (**A**) and scatter plot of KEGG pathways enriched (**B**) with differentially expressed genes (DEGs) detected in the comparison A3_2 vs. A2_2. For Figure 6B, the point size represented the number of DEGs, and colors indicated $p_{adj}$ ranges. $p_{adj} < 0.05$ indicates significant enrichment. The abscissa is the ratio of the number of DEGs annotated with a Kyoto Encyclopedia of Genes and Genomes (KEGG) pathway to the total number of DEGs, and the ordinate is the KEGG pathway.

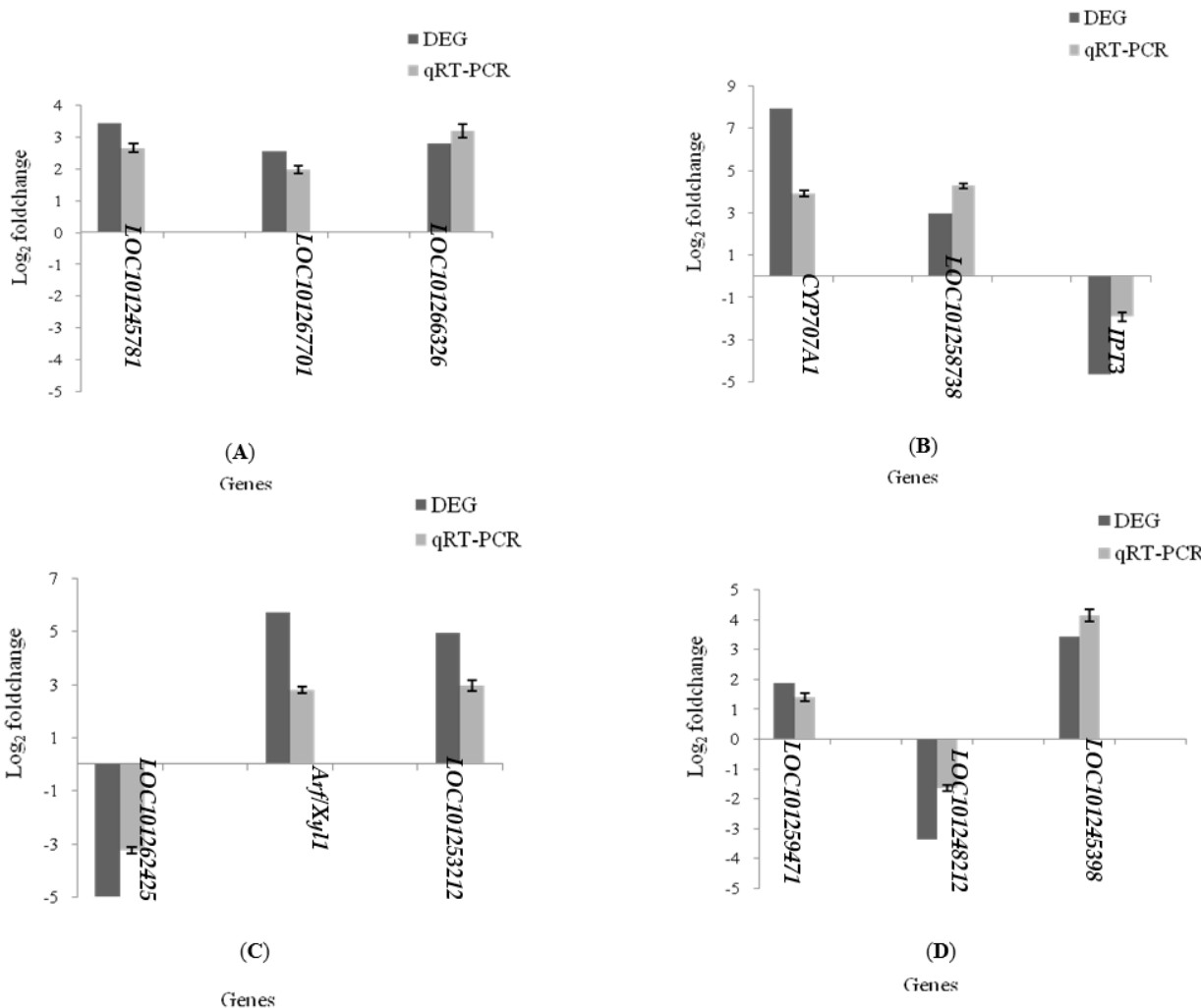

**Figure 7.** Quantitative real-time PCR validation of differential expression genes. (**A**), (**B**), (**C**), and (**D**) indicate the comparisons A3_1 vs. A2_1, A3_2 vs. A2_2, A2_2 vs. A1_2, and A2_1 vs. A1_1, respectively. DEG: differentially expressed gene; qRT-PCR: quantitative real-time PCR.

## 4. Discussion

Cd toxicity can affect plant growth and development by disturbing auxin homeostasis. Researchers have previously discussed the effects of plant hormones under heavy metal stress, including abscisic acid and auxin, and revealed their correlation with stress resistance [33]. Treatment with Cd alone interferes with auxin homeostasis in *Arabidopsis* and disrupts the physiological activities of auxin and lignification in poplar [34,35]. In *Arabidopsis*, the dual toxicity of Cd and arsenic affects the formation and maintenance of post-embryonic roots and interferes with auxin synthesis and transportation [36]. Previous studies have focused on the interference of Cd stress on auxin changes in plants. Compared with wild-type rice plants, the *osaux1* mutant shows decreased auxin content [37]. In the present study, exogenous auxin and the polar auxin transport inhibitor TIBA were applied to plants exposed to Cd stress (Figure 1). The results confirmed that auxin alleviated Cd stress promoted aboveground and belowground biomass accumulation, and alleviated stress damage to a certain extent (Table 1). Auxin positively regulated the resistance of tomato plants to Cd stress mainly by reducing the upward transport of Cd. Moreover, 0.5 μM NAA treatment down-regulated Cd uptake, effectively reduced the upward transport of Cd in tomato plants, and reduced the toxicity of Cd in the shoots (Figure 3). By contrast, application of 1 μM NAA was less effective, which might be related to the inhibition of root growth at a high NAA concentration. This phenomenon was similar to findings

reported by previous studies. *Bacillus amyloliquefaciens* SAY09 improved plant resistance to Cd stress by activating the auxin signaling pathway [38]. Zhu et al. showed that exogenous auxin treatment reduced the upward transport of Cd and restored aboveground growth by accumulating Cd in roots [39].

Previous studies on tomato have shown that exogenous epibrassinolide, polyaspartic acid, and other substances reduce the accumulation of ROS, improve cell wall and vacuole activity, alleviate the toxicity of Cd, and enhance tolerance to Cd stress [5]. The previous studies showed that the proline and MDA content decreases with the increase in Cd and copper concentration [6]. Cd stress damaged the cell membrane structure and caused leakage. The 0.5 μM NAA treatment maintained a relatively intact membrane structure and alleviated membrane lipid peroxidation. By contrast, TIBA treatment promoted this process, which was aggravated with an increase in TIBA concentration (Figure 2A,B). The results supported the conclusion that auxin signaling was involved in the regulation of tomato cell membrane structure under Cd stress. Application of Cd reduces the increased accumulation of $H_2O_2$ and $O_2^-$. *osaux1* is Cd sensitive, and regulation by *osaux1* on the Cd stress response is associated with the metabolism of ROS [37]. Exogenous selenocysteine and melatonin up-regulate the endogenous melatonin content and alleviate the electrolyte exudation and growth inhibition induced by Cd treatment [40]. The present results revealed that application of NAA efficiently ameliorates Cd-induced oxidative stress as characterized by low ROS accumulation, reduced $H_2O_2$ content, and decreased $O_2^-$ production rate (Figure 2C,D). Hasan et al. applied different concentrations of melatonin to tomato plants exposed to Cd stress and reported that the optimal dose of melatonin improves $H^+$-ATPase activity, micronutrient absorption rate, and antioxidant capacity; enhances the synthesis of glutathione and chelating peptide; promotes photosynthesis; increases biomass; and alleviates adverse stress-related effects by accumulating Cd in vacuoles [41]. In the present study, NAA activated the antioxidant system, including SOD, POD, and CAT, which might partially contribute to ROS scavenging under Cd stress (Table 3). It is quite likely that NAA acts as a signal molecule to trigger defense systems, such as the antioxidant system, resulting in the alleviation of oxidative stress. Accordingly, NAA protects the plasma membrane lipid environment by preventing Cd-induced ROS generation and maintaining increased levels of antioxidants. Compared with previous studies, NAA had multiple effects, which played a role in regulating growth and development and improving endogenous material circulation and Cd transfer in plants.

Cd affects the growth of primary roots in *Arabidopsis* by altering *scr* gene expression and simultaneously mediating the auxin–cytokinin pathway [42]. In the present study, the NAA-mediated response to Cd stress in tomato was strongly associated with "defense response" genes in the shoots and genes involved in "oxidoreductase activity, oxidizing metal ions" and "response to auxin" in the roots (Figure 6). KEGG enrichment analysis revealed that Cd stress induced changes in the expression of genes associated with cutin, suberin, and wax biosynthesis in the shoots and with flavonoid biosynthesis in the roots. Flavonoids widely distributed in plants affect auxin transport and negatively regulate auxin transport (Figures S1 and S2). Tobacco transformants overexpressing both serine acetyltransferase and cysteine synthase genes, which are committed to the final two steps of cysteine biosynthesis, exhibit increased resistance to Cd stress compared with that of wild-type and single-gene transgenic plants [43]. Hasan et al. observed that GSH-induced enhancement of Cd stress tolerance is closely associated with upregulation of the transcription of several transcription factors, such as *ETHYLENE RESPONSIVE TRANSCRIPTION FACTOR 1 (ERF1)*, *ERF2*, *MYB1 TRANSCRIPTION FACTOR-AIM 1*, and *R2R3-MYB TRANSCRIPTION FACTOR-AN2*, as well as certain stress-responsive genes [9]. The results indicated that it is feasible to modify auxin homeostasis, activate the auxin signaling pathway, and thereby reduce Cd toxicity and its inhibition of plant root growth and development. However, the underlying mechanism requires further verification by functional identification of the regulatory pathways. The specific mechanism needs to be explored through over-expression and silencing of the key genes using the tomato 'Zhong-

shu No. 6' genetic transformation system established by the research team. Among the differentially expressed genes, tomato metallocarboxypeptidase inhibitor *TCMP*-2 (*2A11*) and *Solanum lycopersicum* heavy metal-associated isoprenylated plant protein (HIPP) 7-like (*LOC101264884*) were specifically expressed in response to NAA treatment under Cd stress. Previous research shows that tomato TCMP-1 which interacts with HIPP26protein was implicated in plant response to cadmium. These two genes are the focus of further functional identification [44]. These genes, which are closely related to plant response to heavy metal stress, may be the key sites of NAA. Meanwhile, it is worth noting that *Solanum lycopersicum* adenylate isopentenyltransferase 3 (*IPT3*) and abscisic acid 8'-hydroxylase CYP707A (*CYP707A*) showed drastic expression changes during this process, which plays an important role in salt or drought stress in plants. *CYP707A* plays an important role in maintaining the homeostasis and balance of endogenous hormones in plants. By changing ABA content, plant dormancy and stomatal closure can be controlled, and stress response can be further regulated [45]. The substantial participation of *SlIPT3* in cytokinins metabolism during salt stress was determined in 35S::*SlIPT3* tomato transformants, where enhancement of cytokinins accumulation significantly improved plant tolerance to salinity [46]. Further research will also focus on the hormone interaction network of these two genes during auxin-mediated cadmium stress.

## 5. Conclusions

In this study, treatment with 0.5 μM NAA improved antioxidant enzyme activities, reduced reactive oxygen, maintained membrane permeability, and decreased MDA and proline contents. The NAA-mediated response to Cd stress was closely associated with "defense response" genes in shoots and "oxidoreductase activity, oxidizing metal ions" and "response to auxin" genes in roots.In further studies, the functions of the two genes, tomato metallocarboxypeptidase inhibitor *TCMP*-2 (*2A11*) and *Solanum lycopersicum* heavy metal-associated isoprenylated plant protein (HIPP) 7-like (*LOC101264884*), will be identified by constructing overexpressed and silenced plants. The potential mechanism of auxin-mediated cadmium tolerance in tomato will be further explored.

**Supplementary Materials:** The following supporting information can be downloaded at: https://www.mdpi.com/article/10.3390/agronomy12092141/s1, Figure S1: Scatter plot of enriched KEGG pathways among genes differentially expressed in the comparison A2_1 vs. A1_1; Figure S2: Scatter plot of enriched KEGG pathways among genes differentially expressed in the comparison A2_2 vs. A1_2; Table S1: RNA-sequencing read quality statistics after filtering; Table S2: Primers used in the study.

**Author Contributions:** X.G., C.S. and K.L. conceived and designed the manuscript. X.G., C.S., Z.C., C.F. and X.D. (Xiufen Dong) analyzed the data and wrote the paper. B.Z., X.D. (Xian Dong) and X.L. reviewed the manuscript. All authors have read and agreed to the published version of the manuscript.

**Funding:** This study was funded by National Natural Science Foundation of China (31672197), Characteristic Laboratory Construction Project of Plant Resources Protection and Application Research in Chishui River Drainage Area (Qian Jiao He KY [2012] 017-4), Guizhou Province Science and Technology Plan Project (QKHZC [2021] No.207), Talent Base for Environmental Protection and Mountain Agricultural in Chishui River Basin, Cultivation Project for Innovative Exploration of Academic New Seedlings in Guizhou Province (Qian Ke He Pingtai Rencai [2017] 5727-18), and Zunyi City Joint Science and Technology Research and Development Fund Project (Zun Shi Ke He HZ [2020] No. 9).

**Institutional Review Board Statement:** Not applicable.

**Informed Consent Statement:** Not applicable.

**Data Availability Statement:** All data, models, and code generated or used during the study appear in the submitted article.

**Conflicts of Interest:** The authors declare no conflict of interest.

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
