# Peer review of "Effects of α-Naphthylacetic Acid on Cadmium Stress and Related Factors of Tomato by Regulation of Gene Expression"

_agronomy, doi:10.3390/agronomy12092141_

Round 1

Reviewer 1 Report

Thank you for this interesting research work. The manuscript is well structured and should be considered for publication in Agronomy after some major revisions. Please check my following comments:

1. Why the authors selected the concentrations of 0.5 and 1 µM/L NAA for the screening, not lower or higher? Please explain.

2. Similarly, why 50 and 100 mg/L TIBA? Please explain.

3. What is the volume of NAA and TIBA solutions the authors used for foliar spray? (How many mL?  µL?)

4. Why the authors selected “Zhongshu No. 6” in this study? Not others? Please explain in the discussion part.

5. Why the authors selected Actin as internal control in this study, not others? There are numerous potential housekeeping genes that are used as a control in quantitative PCR. The expression stability of the reference gene could vary based on the tissue types, the developmental stages, and the experimental conditions (e.g., Cd stress in this case). The expression stability of various housekeeping genes under stress and control conditions should be tested and the housekeeping gene with the most stable expression under test (stress) and control conditions should be selected as a reference housekeeping gene. Ideally, the authors should calculate inter-group variance and intra-group variance for both (test & control), and the result should be plotted and included. Refer to this study (https://doi.org/10.1111/j.1744-7348.2011.00506.x). If the authors also tested for other housekeeping genes, please present the results in Supplementary Data. If not, please clearly indicate the reason why the authors selected Actin in the discussion part.

6. What is the total sample size (n) for ANOVA analysis?

7. In the results of Auxin distribution (Section 3.1), why only Cd+N1 and Cd+T1 outcomes were presented? Why Cd+N2 and Cd+T2 were not indicated? Please explain.

8. Data in Table 1 are not well presented, especially in the columns of fresh and dry weight. Please revise. No need the word “Note” in the table footer, also for other tables. 

9. Also, please indicate clearly, for example, data are expressed as mean ± SD (or SE?), followed by capital letters and lowercase letters. Otherwise, capital letters can be misunderstood with the letter “C” of treatment (Control).

10. Please revise Figure 2 which is not well presented. Please clearly separate the legends on the horizontal axis. Also, arrange the grouping letters more appropriately. 

11. Please increase the performance of Figure 4A, the letters are too small and not clear.

12. Please explain what are A, B-1, B-2, etc. in Figure 5 caption. Also, what are A1_2, A2_1, A1_1, etc. in the horizontal axis? Otherwise, hard to understand.

13. What are DEG, qRT-… in Figure 7? Please explain them in the figure caption.

14. Overall, along with the interesting results, the discussion part is too general and too weak. The authors should discuss more deeply and broadly. For example, the possible mechanisms or the signal pathways by which NAA exogenous application can increase tomato growth performance, chlorophyll content, antioxidant activity, antioxidant enzymes, etc. under Cd stress. The results should be discussed and compared part by part in detail with that of other studies, not in a very general way in the present form. Discussion about research design and applied methods should also be improved.

Author Response

Dear Editors and Reviewers:

Thank you for your letter and for the reviewers’ comments concerning our manuscript entitled ”Effects of α-Naphthylacetic Acid on Cadmium Stress and Related Factors of Tomato by Regulation of Gene Expression” (ID: agronomy-1895161). Those comments are all valuable and very helpful for revising and improving our paper, as well as the important guiding significance to our researches. We have studied comments carefully and have made correction which we hope meet with approval. Revised portion are marked in red in the paper. The main corrections in the paper and the responds to the reviewer’s comments are as flowing:

  1. It is really true as Reviewer suggested that the spray concentration in this study is of great importance. First, the duality of auxin directly determines this point: it can promote plant growth in the low concentration range, inhibit plant growth in the high concentration range, and even kill plants. Second, different organs have different sensitivity to auxin.For the same plant, the sensitivity of auxin is usually: root > bud > stem. How to confirm the appropriate concentration, so as to maximize the effect of auxin in polar transportation and distribution to various organs,which has also become the key of this study. In addition, natural auxin can maintain a dynamic balance between synthesis and decomposition in plants. There is an essential difference between auxin analogues and natural auxin, especially naphthylacetic acid, which has a long-term and stable effect. It can not be quickly metabolized by the plant itself if applied too much, and can not be quickly synthesized by the plant itself if applied too little. Through a large number of pre experiments, the optimal spraying concentration of NAA solution was confirmed to be 0.5 and 1μM.
  2. TIBA is the specific inhibitor of auxin to block the polar transport of auxin. The application of TIBA in this study focuses on the reverse validation of the role of NAA. Through pre experiments, the optimal spraying concentration of TIBA solution was determined to be 50 and 100 mg/L (Converted by molecular weight to 100 and 200 μM).
  3. The operation is to ensure that both the leaf surface and back were sprayed, and the dosage is about 2ml per plant.
  4. This part has been stated in the discussion. The specific mechanism needs to be explored through over-expression and silencing of the key genes using the tomato ‘Zhongshu No. 6’ genetic transformation system established by the research team. In further studies, the functions of the two genes, tomato metallocarboxypeptidase inhibitor TCMP-2 (2A11) and Solanum lycopersicum heavy metal-associated isoprenylated plant protein (HIPP) 7-like (LOC101264884) will be identified by constructing overexpressed and silenced plants. The potential mechanism of auxin mediated cadmium tolerance in tomato will be further explored.
  5. Actin is an housekeeping gene that the research team found to be stable in tomato plants under various conditions [1, 2]. qRT-PCR of DEGs was performed to verify RNA-Seq data. The results were consistent with those of RNA-Seq analysis and confirmed that the RNA-seq data were reliable.

[1] Guan X, Xu T, Gao S, et al. Temporal and Spatial Distribution of Auxin Response Factor Genes During Tomato Flower Abscission. J Plant Growth Regul, 2014, 33(2): 317-327

[2] Xu T, Liu X, Wang R, et al. SlARF2a plays a negative role in mediating axillary shoot formation. Scientific Reports, 2016, 6: 33728

  1. In this study, n=3 for ANOVA analysis.
  2. This section does not focus on the comparison of different concentrations of NAA and TIBA. The polar transport of auxin is from the upper end of morphology to the lower end of morphology. The two treatments of Cd+N1 and Cd+T1 were selected only to intuitively show the effect of NAA and TIBA application on the distribution of IAA in plants, so as to assist the following description of the alleviating effect of NAA on cadmium stress.
  3. Considering the Reviewer’s suggestion, we divide Table 1 into two tables: Table 1 and Table 2. The word “Note” in the table footer were all removed.
  4. As Reviewer suggested that we only keep the 5% level and present it in the above corner.
  5. Considering the Reviewer’s suggestion, Figure 2 has been adjusted.
  6. Considering the Reviewer’s suggestion, the performance of Figure 4A has been adjusted.
  7. In ‘3.6’, it is explained as follows: The combined gene expression patterns were grouped into four major clusters, which were designated Clusters A, B, C, and D (Figure 5). The differential expression of genes in Clusters A, B, and C was mainly concentrated in the roots. The majority of Cluster A genes were up-regulated under Cd treatment and down-regulated under Cd+N1 treatment. Cluster B-1 genes were down-regulated under Cd treatment. Cluster B-2 genes were up-regulated under Cd treatment and down-regulated under Cd+N1 treatment. Cluster C-1 genes were down-regulated under Cd treatment and up-regulated under Cd+N1 treatment. Cluster C-2 genes were down-regulated under Cd and Cd+NAA treatments. Cluster D-1 genes were up-regulated under Cd treatment and down-regulated slightly under Cd+N1 treatment. Cluster D-2 genes were down-regulated under Cd treatment and up-regulated under Cd+N1 treatment.
  8. It is really true as Reviewer suggested that we have explain DEG and qRT-…in the figure caption. DEG: Differentially expressed gene, qRT-…: Quantitative Real-time PCR
  9. We have re-written the discussion part from P14 to P16 according to the Reviewer’s suggestion.

Special thanks to you for your good comments.We tried our best to improve the manuscript and made some changes in the manuscript. These changes will not influence the content and framework of the paper. And here we did not list the changes but marked in red in revised paper.We appreciate for Editors/Reviewers’ warm work earnestly, and hope that the correction will meet with approval.

Reviewer 2 Report

The manuscript to Guan et al. by "Effects of Naphthylacetic Acid on Cadmium Stress and Related Factors of Tomato through Differential Gene Expression" (agronomy-1895161), demonstrated the effects of Cadmium and interaction of NAA and TIBA (synthetic auxin and inhibidor, respectively) in response a transgenic tomato with GUS reporter gene.

The authors have done a large amount of work, employing various references and critical analysis based on a scientific method and structure. The introduction its contextuaized and Ok. However, many section, its necessary adjusts. Material and methods its necessary improve some rephrase and reference methods. The results section needs revision. The comparisons described are often confusing. Many discussion phrases are present in the results section. The discussion needs to be improved. For example, the enzymatic data reported (CAT, POD, SOD) is not discussed in manuscript. The transcriptome data have rarely been discussed in relation to the context of cadmium in tomato. As there is no number of lines in the manuscript, I attached the pdf with suggestions for changes and phrases which I did not understand.

Figures and tables need minor adjustments or reformulation.

However, it is an interesting manuscript, with robust data and that fits the journal's scopus, requiring modifications only in the manuscript. I think that after the major modifications, it can be accepted for publication.

In addition, a attached a pdf file, with major corrections to English Grammer, suggesting and questions which its not clearance for me. Please, check in manuscript American or British English.

Best regards

Author Response

Dear Editors and Reviewers:

Thank you for your letter and for the reviewers’ comments concerning our manuscript entitled ”Effects of α-Naphthylacetic Acid on Cadmium Stress and Related Factors of Tomato by Regulation of Gene Expression” (ID: agronomy-1895161). Those comments are all valuable and very helpful for revising and improving our paper, as well as the important guiding significance to our researches. We have studied comments carefully and have made correction which we hope meet with approval. Revised portion are marked in red in the paper. The main corrections in the paper and the responds to the reviewer’s comments are as flowing:

  1. We have made correction according to the Reviewer’s comments. The section of ‘material and methods’ and ‘results’ have been improved according to the PDF attached. For deep and systematic discussion, some phrases in the results section have been moved to the discussion section.
  2. It is really true as Reviewer suggested that the discussion should been improved. According to the suggestions made by reviewer, we specially rewrote a paragraph about ROS scavenging mechanisms of AOE (SOD, POD, and CAT) owing to Cd stress. Formulation of this part is as follows:

Previous studies on tomato have shown that exogenous epibrassinolide, polyaspartic acid, and other substances reduce the accumulation of ROS, improve cell wall and vacuole activity, alleviate the toxicity of Cd, and enhance tolerance to Cd stress [5]. The previous studies showed that the proline and MDA content decreases with the increase in Cd and copper concentration [6]. Cd stress damaged the cell membrane structure and caused leakage. The 0.5 μM NAA treatment maintained a relatively intact membrane structure and alleviated membrane lipid peroxidation. By contrast, TIBA treatment promoted this process, which was aggravated with an increase in TIBA concentration (Figure 2A, B). The results supported the conclusion that auxin signaling was involved in the regulation of tomato cell membrane structure under Cd stress. Application of Cd reduces the increased accumulation of H2O2 and O2–.. osaux1 is Cd sensitive, and regulation by osaux1 on the Cd stress response is associated with the metabolism of ROS [37]. Exogenous selenocysteine and melatonin up-regulate the endogenous melatonin content and alleviate the electrolyte exudation and growth inhibition induced by Cd treatment [40]. The present results revealed that application of NAA efficiently ameliorate Cd-induced oxidative stress as characterized by low ROS accumulation, reduced H2O2 content and decreased O2−· production rate (Figure 2C, D). Hasan et al. applied different concentrations of melatonin to tomato plants exposed to Cd stress and reported that the optimal dose of melatonin improves H+-ATPase activity, micronutrient absorption rate, and antioxidant capacity; enhances the synthesis of glutathione and chelating peptide; promotes photosynthesis; increases biomass; and alleviates adverse stress-related effects by accumulating Cd in vacuoles [41]. In the present study, NAA activated the antioxidant system, including SOD, POD and CAT, which might partially contribute to ROS scavenging under Cd stress (Table 3). It is quite likely that NAA acts as a signal molecule to trigger defense systems, such as the antioxidant system, resulting in the alleviation of oxidative stress. Accordingly, NAA protects the plasma membrane lipid environment by preventing Cd-induced ROS generation and maintaining increased levels of antioxidants. Compared with previous studies, NAA had multiple effects, which played a role in regulating growth and development and improving endogenous material circulation and Cd transfer in plants.

  1. Considering the Reviewer’s suggestion, we added comprehensive analysis of key genes screened by transcriptome sequencing in the discussion. Formulation of this part is as follows:

The specific mechanism needs to be explored through over-expression and silencing of the key genes using the tomato ‘Zhongshu No. 6’ genetic transformation system established by the research team. Among the differentially expressed genes, tomato metallocarboxypeptidase inhibitor TCMP-2 (2A11) and Solanum lycopersicum heavy metal-associated isoprenylated plant protein (HIPP) 7-like (LOC101264884) were specifically expressed in response to NAA treatment under Cd stress. Previous research shows that tomato TCMP-1 which interacts with HIPP26 protein was implicated in plant response to cadmium These two genes are the focus of further functional identification [44]. These genes, which are closely related to plant response to heavy metal stress, may be the key sites of NAA. Meanwhile, it is worth noting that Solanum lycopersicum adenylate isopentenyltransferase 3 (IPT3) and abscisic acid 8'-hydroxylase CYP707A (CYP707A) showed drastic expression changes during this process, which have been proved to play an important role salt or drought stress in plant. CYP707A plays an important role in maintaining the homeostasis and balance of endogenous hormones in plants. By changing ABA content, plant dormancy and stomatal closure can be controlled, and stress response can be further regulated [45]. The substantial participation of SlIPT3 in cytokinins metabolism during salt stress has been determined in 35S::SlIPT3 tomato transformants, where enhancement of cytokinins accumulation significantly improved plant tolerance to salinity [46]. Further research will also focus on the hormone interaction network of these two genes during auxin mediated cadmium stress.

  1. We have made correction according to the Reviewer’s PDF, including English Grammer, figures and tables.

Special thanks to you for your good comments. We tried our best to improve the manuscript and made some changes in the manuscript. These changes will not influence the content and framework of the paper. And here we did not list the changes but marked in red in revised paper. We appreciate for Editors/Reviewers’ warm work earnestly, and hope that the correction will meet with approval.

Reviewer 3 Report

The manuscript ‘Effects of Naphthylacetic Acid on Cadmium Stress and Related Factors of Tomato through Differential Gene Expression by Guan et al. describes the role of NAA and TIBA on Cd stress tolerance in tomato. The manuscript is well-planned, well-executed, and well-written. I have a few observations on the manuscript which may be useful for further improvement;

Ø In the title the authors have mentioned ‘Naphthylacetic Acid’-chemical formula: C12H10O2 but it the text, they are talking about ‘α-naphthaleneacetic acid’-chemical formula: C10H7CH2CO2H. Please confirm and use the right term ‘α-naphthaleneacetic acid’ throughout the manuscript.

Ø Please discuss about some differentially expressed genes (DEGs) in the abstract.

Ø Please include Solanum lycopersicum, α-naphthaleneacetic acid, Cadmium stress, gene expression.

Ø Author may consider introducing a line on antioxidative enzyme (AOE) mechanisms on Cd stress tolerance following AsA-GSH cycle.

Ø ROS and lignin deposition (H2O2 and MDA) may be introduced first followed by the scavenging mechanism, AOE, and gene expression in the introduction.

Ø In the section 2.2; To explore the effect of auxin, four treatments (S, Cd, Cd+N1, and Cd+T1) were stained with the GUS. Why the authors have not included Cd+N2, and Cd+T2???? Pl justify.

Ø In the section 2.3; Authors may please include the enzyme catalogue number (EC number) for SOD, POD, and CAT.

Ø In the section 2.5; Please mention about the biological replicates used for RNA-Seq analysis.

Ø Results are well-explained.

Ø In the tables, the authors presented the level of significance at 5% and 1%, which is not required. They can present the level of significance at 5% or 1%. Accordingly, alphabets denoting the level of significance may be placed after the values.

Ø Figure 3: the titles of Y, and Y’ axis may be increased to a visible size.

Ø Similarly, the title of X, and Y axis in Figure 4, and gene numbers in the Venn diagram may be increased.

Ø  Figure 4 and Figure 7 each may be presented in a compact panel (A-D).

Ø The results were discussed properly. However, I wish to see a paragraph of discussion on ROS scavenging mechanisms of AOE (SOD, POD, and CAT) owing to Cd stress.

Ø In conclusion, please write a few points on future implication of the study. Such as developing Cd tolerant lines involving highly up-regulated lines or knocking down the key down-regulated genes.

Overall, the manuscript is a well-written, may be considered for publication with minor revision.

Good luck with the revision.

Author Response

Dear Editors and Reviewers:

Thank you for your letter and for the reviewers’ comments concerning our manuscript entitled ”Effects of α-Naphthylacetic Acid on Cadmium Stress and Related Factors of Tomato by Regulation of Gene Expression” (ID: agronomy-1895161). Those comments are all valuable and very helpful for revising and improving our paper, as well as the important guiding significance to our researches. We have studied comments carefully and have made correction which we hope meet with approval. Revised portion are marked in red in the paper. The main corrections in the paper and the responds to the reviewer’s comments are as flowing:

  1. We are very sorry for our negligence of the title. The title was corrected to: Effects of α-Naphthylacetic Acid on Cadmium Stress and Re-lated Factors of Tomato by Regulation of Gene Expression.
  2. Considering the Reviewer’s suggestion, we discuss about the key DEGs in the abstract. Formulation of this part is as follows:

Among the differentially expressed genes, tomato metallocarboxypeptidase inhibitor TCMP-2 (2A11) and Solanum lycopersicum heavy metal-associated isoprenylated plant protein (HIPP) 7-like (LOC101264884) , which are closely related to plant response to heavy metal stress, may be the key sites of NAA.

  1. It is really true as Reviewer suggested that the keywords of the manuscript include Solanum lycopersicum, α-naphthaleneacetic acid, Cadmium stress, gene expression.
  2. Considering the Reviewer’s suggestion, we have added the introduction of antioxidative enzyme (AOE) mechanisms on Cd stress tolerance following AsA-GSH cycle. Formulation of this part is as follows:

Under suitable conditions, the antioxidant enzyme system can eliminate the reactive oxygen radicals induced by stress, and alleviate the damage caused by stress to a certain extent. SOD firstly disproportionates reactive oxygen radicals and then converts them into H2O2 which can be cleared by CAT and POD.

  1. We have added this part according to the Reviewer’s suggestion. Formulation of this part is as follows:

Various environmental conditions will lead to a large number of reactive oxygen species and free radicals in plants. H2O2 and O2–. are the most important reactive oxygen species in plants, which will lead to membrane lipid peroxidation and indirectly reflect the degree of stress. Studies have shown that many foreign substances can reduce the accumulation of ROS in plants and enhance tolerance to Cd stress [5]. Proline is an important osmoregulatory substance that is readily induced by stress. As one of the most important products of membrane lipid peroxidation, MDA is a marker of cell membrane system damage [6].

  1. This section does not focus on the comparison of different concentrations of NAA and TIBA. The polar transport of auxin is from the upper end of morphology to the lower end of morphology. The two treatments of Cd+N1 and Cd+T1 were selected only to intuitively show the effect of NAA and TIBA application on the distribution of IAA in plants, so as to assist the following description of the alleviating effect of NAA on cadmium stress.
  2. As Reviewer suggested that we have the enzyme catalogue number (EC number) for SOD: EC1.15.1.1, POD: EC 1.11.1.7, CAT: EC 1.11.1.6.
  3. Each treatment had 3 biological replicates for RNA-Seq analysis.
  4. As Reviewer suggested that we only keep the 5% level and present it in the above corner.
  5. Considering the Reviewer’s suggestion, Figure 3 has been adjusted.
  6. Considering the Reviewer’s suggestion, Figure 4 has been adjusted.
  7. Figure 4 and Figure 7 will be subject to editing arrangement in later typesetting.
  8. It is really true as Reviewer suggested that the discussion should been improved. According to the suggestions made by reviewer, we specially rewrote a paragraph about ROS scavenging mechanisms of AOE (SOD, POD, and CAT) owing to Cd stress. Formulation of this part is as follows:

Previous studies on tomato have shown that exogenous epibrassinolide, polyaspartic acid, and other substances reduce the accumulation of ROS, improve cell wall and vacuole activity, alleviate the toxicity of Cd, and enhance tolerance to Cd stress [5]. The previous studies showed that the proline and MDA content decreases with the increase in Cd and copper concentration [6]. Cd stress damaged the cell membrane structure and caused leakage. The 0.5 μM NAA treatment maintained a relatively intact membrane structure and alleviated membrane lipid peroxidation. By contrast, TIBA treatment promoted this process, which was aggravated with an increase in TIBA concentration (Figure 2A, B). The results supported the conclusion that auxin signaling was involved in the regulation of tomato cell membrane structure under Cd stress. Application of Cd reduces the increased accumulation of H2O2 and O2–.. osaux1 is Cd sensitive, and regulation by osaux1 on the Cd stress response is associated with the metabolism of ROS [37]. Exogenous selenocysteine and melatonin up-regulate the endogenous melatonin content and alleviate the electrolyte exudation and growth inhibition induced by Cd treatment [40]. The present results revealed that application of NAA efficiently ameliorate Cd-induced oxidative stress as characterized by low ROS accumulation, reduced H2O2 content and decreased O2−· production rate (Figure 2C, D). Hasan et al. applied different concentrations of melatonin to tomato plants exposed to Cd stress and reported that the optimal dose of melatonin improves H+-ATPase activity, micronutrient absorption rate, and antioxidant capacity; enhances the synthesis of glutathione and chelating peptide; promotes photosynthesis; increases biomass; and alleviates adverse stress-related effects by accumulating Cd in vacuoles [41]. In the present study, NAA activated the antioxidant system, including SOD, POD and CAT, which might partially contribute to ROS scavenging under Cd stress (Table 3). It is quite likely that NAA acts as a signal molecule to trigger defense systems, such as the antioxidant system, resulting in the alleviation of oxidative stress. Accordingly, NAA protects the plasma membrane lipid environment by preventing Cd-induced ROS generation and maintaining increased levels of antioxidants. Compared with previous studies, NAA had multiple effects, which played a role in regulating growth and development and improving endogenous material circulation and Cd transfer in plants.

  1. 1 Considering the Reviewer’s suggestion, we have added a few points on future implication of the study in conclusion. Formulation of this part is as follows:

In further studies, the functions of the two genes, tomato metallocarboxypeptidase inhibitor TCMP-2 (2A11) and Solanum lycopersicum heavy metal-associated isoprenylated plant protein (HIPP) 7-like (LOC101264884) will be identified by constructing overexpressed and silenced plants. The potential mechanism of auxin mediated cadmium tolerance in tomato will be further explored.

Special thanks to you for your good comments. We tried our best to improve the manuscript and made some changes in the manuscript. These changes will not influence the content and framework of the paper. And here we did not list the changes but marked in red in revised paper. We appreciate for Editors/Reviewers’ warm work earnestly, and hope that the correction will meet with approval.

Round 2

Reviewer 1 Report

I highly appreciate the efforts of the authors to revise their manuscript.

The updated version has been significantly improved. However, the performance of Tables and Figures is still limited. In my opinion, they will be more attractive if the authors can improve their quality. Please find my comments as below:

1.     Please re-format the Table footers and Figure captions following the MDPI format.

2.     Please also include the footer for Table 2.

3.     I understand that the abbreviations of sample codes are explained in the text. But in my opinion, it should be presented also in Table footers and Figure captions. Based on that, readers can easily understand the results shown in Tables and Figures without looking for the explanation in the main text. So, please also include this point in Table 3, Figures 3 and 5.

4.     Figure 7 is still presented in low quality. Figure 7A: Gene names on x-axis are not well separated, difficult to distinguish. Figure 7B and 7D: Gene names on x-axis are overlapped with columns and y-axis. Figure 7C: Gene names on x-axis are overlapped with y-axis. Difficult to read. In addition, it should be mentioned as “qRT-PCR” instead of “qRT-…”.

5.     The authors seem to be not familiar with using MDPI journal template. I suggest the author should look for the instruction to use this. It will help the authors improve the manuscript’s performance.

After addressing the above-mentioned limitations, I suggest that the manuscript can be considered to be accepted.

Author Response

Dear Editors and Reviewers:

Thank you for your letter and for the reviewers’ comments concerning our manuscript entitled ”Effects of α-Naphthylacetic Acid on Cadmium Stress and Related Factors of Tomato by Regulation of Gene Expression” (ID: agronomy-1895161). Those comments are all valuable and very helpful for revising and improving our paper, as well as the important guiding significance to our researches. We have studied comments carefully and have made correction which we hope meet with approval. Revised portion are marked in red in the paper. The main corrections in the paper and the responds to the reviewer’s comments are as flowing:

  1. Considering the Reviewer’s suggestion, we have perfected Table footers and Figure captions following the MDPI format.
  2. Considering the Reviewer’s suggestion, the footer for Table 2 have been added.
  3. It is really true as Reviewer suggested that the abbreviations of sample codes have been explained in Table footers and Figure captions include Table 3, Figures 3 and 5.
  4. Considering the Reviewer’s suggestion, Figure 7 has been replaced.
  5. Considering the Reviewer’s suggestion, We have improved the manuscript’s performance following the MDPI journal template.

Special thanks to you for your good comments. We tried our best to improve the manuscript and made some changes in the manuscript. These changes will not influence the content and framework of the paper. And here we did not list the changes but marked in red in revised paper. We appreciate for Editors/Reviewers’ warm work earnestly, and hope that the correction will meet with approval.

Reviewer 2 Report

Dear Authors,

The paper has been improved greatly after the revision. Thank you for addressing all my comments. In addition, the structure is adequate, the information is new and of great significance for comprehension. I recommend the acceptance of revised manuscript.

Best regards.

Author Response

Dear Editors and Reviewers:

Thank you for your letter and for the reviewers’ comments concerning our manuscript entitled ”Effects of α-Naphthylacetic Acid on Cadmium Stress and Related Factors of Tomato by Regulation of Gene Expression” (ID: agronomy-1895161). Those comments are all valuable and very helpful for revising and improving our paper, as well as the important guiding significance to our researches.

Special thanks to you for your good comments. We appreciate for Editors/Reviewers’ warm work earnestlyl.